# EQUIVARIANT NORMALIZING FLOWS
# FOR POINT PROCESSES AND SETS

## ABSTRACT

A point process describes how random sets of exchangeable points are generated. The points usually influence the positions of each other via attractive and repulsive forces. To model this behavior, it is enough to transform the samples from the uniform process with a sufficiently complex equivariant function. However, learning the parameters of the resulting process is challenging since the likelihood is hard to estimate and often intractable. This leads us to our proposed model – CONFET. Based on continuous normalizing flows, it allows arbitrary interactions between points while having tractable likelihood. Experiments on various real and synthetic datasets show the improved performance of our new scalable approach.

## 1 INTRODUCTION

Many domains contain unordered data with a variable number of elements. The lack of ordering, also known as exchangeability, can be found in locations of cellular stations, locations of trees in a forest, point clouds, items in a shopping cart etc. This kind of data is represented with sets that are randomly generated from some underlying process that we wish to uncover. We choose to model this with spatial point processes, generative models whose realizations are sets of points.

Perhaps the simplest non-trivial model is an inhomogeneous Poisson process. The locations of the points are assumed to be generated i.i.d. from some density (Chiu et al., 2013). By simply modeling this density we can evaluate the likelihood and draw samples. We can do this easily with normalizing flows (Germain et al., 2015). The process is then defined with a transformation of samples from a simple distribution to samples in the target distribution. However, the i.i.d. property is often wrong because the presence of one object will influence the distribution of the others. For example, short trees grow near each other, but are inhibited by taller trees (Ogata & Tanemura, 1985).

As an example we can generate points on the interval $(0, 1)$ in the following way: we first flip a coin to decide whether to sample inside or outside of the interval $(1/4, 3/4)$; then sample two points $x_1$, $x_2$ uniformly on the chosen subset. Although the marginals $p(x_1)$ and $p(x_2)$ are uniformly distributed, knowing the position of one point gives us information about the other. Therefore, we should not model this process as if the points are independent, but model the joint distribution $p(x_1, x_2)$, with a constraint that $p$ is symmetric to permutations (Figure 1). Unfortunately, when we include interactions between the points, the problem becomes significantly harder because the likelihood is intractable (Besag, 1975).

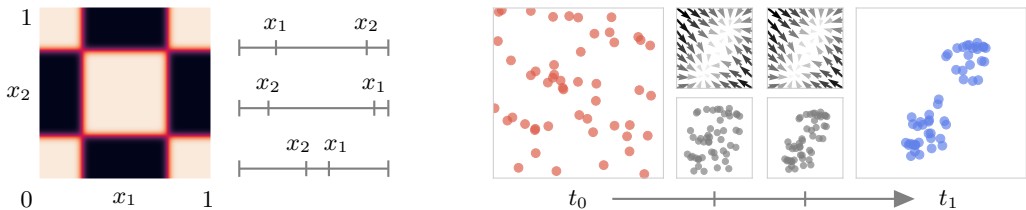

Figure 1: (Left) A symmetric density and its samples, see details in text. (Right) Illustration of our approach, going from the uniform to the target process with a continuous normalizing flow.

In this work, we present a way to solve this issue by using continuous normalizing flows that allow unrestricted transformations of points, with interactions between them. This way, a very hard problem of likelihood estimation suddenly becomes tractable. Our approach transforms simple processes into complex ones, by transforming their samples with expressive functions (Figure 1). Our main contributions are the following:

- We reinterpret and unify existing techniques for modeling point processes and exchangeable data with normalizing flows.
- We propose a new generative model (CONFET) that allows interactions between points and can be trained with maximum likelihood. The extensive experiments show that it outperforms other approaches, while remaining efficient and scalable.

## 2 POINT PROCESSES AND RANDOM SETS

Realizations of a finite point process on a bounded region $B \subset \mathbb{R}^d$ are finite sets of points $X = \{\boldsymbol{x}_1, \ldots, \boldsymbol{x}_n\}$, $\boldsymbol{x}_i \in B$. A point process is simple if no two points fall at exactly the same place. In practice, point processes are usually both finite and simple.

One way to construct a general point process is by defining a discrete distribution $p(n)$ for the number of points and a symmetric probability density $p(X)$ on $B^n$ for their locations (Daley & Vere-Jones, 2007). The symmetry requirement comes from the fact that the probability of a sequence $(\boldsymbol{x}_{\pi(1)}, \ldots, \boldsymbol{x}_{\pi(n)})$ is the same for any permutation of the elements $\pi$ — the points are exchangeable.

Not knowing the order of the points can be solved trivially by averaging any density over all $n!$ permutations. Another approach is to impose a canonical order, e.g. sorting the points by one of the dimensions. Then the probability of observing a set of exchangeable elements is defined w.r.t. the joint probability of an order statistic $\boldsymbol{x}_{(1)} < \cdots < \boldsymbol{x}_{(n)}$ (Casella & Berger, 2002):

$$p(X) = \frac{1}{n!} p(\boldsymbol{x}_{(1)}, \ldots, \boldsymbol{x}_{(n)}). \tag{1}$$

A traditional way to define a point process is with an intensity function that assigns a non-negative value to every subset of $B$, corresponding to a number of points we expect to see there (Møller & Waagepetersen, 2003). An example is a *homogeneous* Poisson process with constant intensity $\lambda$. To generate a new realization, we first sample $n \sim \text{Pois}(\lambda)$, and then sample $n$ points uniformly on $B$.

If we define the intensity as a function of position $\lambda(\boldsymbol{x})$, we get an *inhomogeneous* Poisson process which is equivalent to defining the un-normalized probability density function on $B$. Now $n$ follows $\text{Pois}(\Lambda)$, where $\Lambda$ is the total intensity $\int \lambda(\boldsymbol{x}) d\boldsymbol{x}$. We get the density at a location $\boldsymbol{x}$ by normalizing the intensity $p(\boldsymbol{x}) = \lambda(\boldsymbol{x})/\Lambda$. Combining the distribution of the number of points with the distribution of their locations gives us a well known formula for the likelihood of an inhomogeneous Poisson process (Daley & Vere-Jones, 2007, eq. 7.1.2):

$$L(X) = \left( \prod_{\boldsymbol{x}_i \in X} \lambda(\boldsymbol{x}_i) \right) \exp\left( -\int_B \lambda(\boldsymbol{x}) d\boldsymbol{x} \right). \tag{2}$$

Instead of modeling $\lambda(\boldsymbol{x})$, we can model $p(\boldsymbol{x})$ directly to avoid estimating the integral, without losing generality (Yuan et al., 2020). This shift in the perspective from intensity to probability density function allows us to utilize rich existing methods from density estimation.

An extension of an inhomogeneous process that allows interactions between points defines the conditional intensity $\lambda(\boldsymbol{x}_i|X)$ (Papangelou, 1974). This may be interpreted as the conditional probability of having a point at $\boldsymbol{x}_i$ given the rest of the process coincides with $X$. The likelihood is not tractable anymore so previous works used pseudolikelihood instead, replacing $\lambda(\boldsymbol{x})$ with $\lambda(\boldsymbol{x}|X)$ in Eq. 2 (Besag, 1975; Baddeley & Turner, 2000).

One example of such a process is a clustering process (Neyman & Scott, 1958) that generates the points in two steps. First, we sample the cluster centers from a Poisson process, then we sample the final points from normal distributions centered around cluster positions. In contrast, a repulsion process (Matérn, 2013) generates initial points from a uniform process and removes those that have neighbors inside radius $R$.

A different perspective on point processes is viewing them as random sets (Baddeley et al., 2006). Recent approaches for modeling sets (Korshunova et al., 2018; Bender et al., 2020) invoke de Finetti's theorem (De Finetti, 1937) which states that the probability of an infinite exchangeable sequence is a mixture of i.i.d. processes:

$$p(X) = \int p(\boldsymbol{z}) \prod_{\boldsymbol{x}_i \in X} p(\boldsymbol{x}_i | \boldsymbol{z}) d\boldsymbol{z}, \tag{3}$$

here written for a finite subset (O'Neill, 2009, Theorem 1). We can use Eq. 3 to construct a latent variable model where points are independent of each other given $\boldsymbol{z}$. Since the integral is intractable, we will have to resort to approximate inference.

Equations 1, 2 and 3 give us different ways to obtain the parameters $\boldsymbol{\theta}$ of a point process by maximizing the likelihood $p_{\boldsymbol{\theta}}(X)$. However, the approaches that arise from them are equivalent in the sense they all model the same symmetric density. Therefore, omitting terms related to cardinality (Vo et al., 2018), like $p(n)$, does not matter when comparing them. Further discussion is in Appendix A.1. Our goal is to have a generative model defined with $\boldsymbol{\theta}$ that can produce sets as realizations. One way to generate realizations of different point processes is to transform the points drawn from a uniform Poisson process with an invertible function. The only requirement is that the process remains locally finite (Baddeley et al., 2006, Section 1.7). To keep track of the likelihood we will use the change of variables formula from the normalizing flow framework.

## 3 UNIFYING POINT PROCESS MODELS WITH NORMALIZING FLOWS

In the following section we review the models for point processes and random sets that use likelihoods listed in Section 2. These models can be found fully or partially in the literature on point processes and modeling exchangeable data. We unify them here under the common umbrella of normalizing flows. Their limitations form the foundation for our proposed solution in Section 4.

A normalizing flow is a generative model that defines a complex distribution as a series of invertible smooth transformations of the initial random variable (Germain et al., 2015). That means, if we apply a function $f : \mathbb{R}^d \to \mathbb{R}^d$ to the random variable $\boldsymbol{z} \sim q(\boldsymbol{z})$, where $f$ is invertible and differentiable, we can get the log-density of $\boldsymbol{x} = f(\boldsymbol{z})$ by calculating the change of variables formula:

$$\log p(\boldsymbol{x}) = \log q(\boldsymbol{z}) - \log \left| \det \frac{\partial f(\boldsymbol{z})}{\partial \boldsymbol{z}} \right| = \log q(f^{-1}(\boldsymbol{x})) + \log \left| \det \frac{\partial f^{-1}(\boldsymbol{x})}{\partial \boldsymbol{x}} \right|. \tag{4}$$

To sample from $p(\boldsymbol{x})$ we first sample $\boldsymbol{z}$ from $q(\boldsymbol{z})$, then apply the forward transformation $\boldsymbol{z} \mapsto \boldsymbol{x}$. To estimate density $p(\boldsymbol{x})$ for a given sample, we apply the inverse transformation $\boldsymbol{x} \mapsto \boldsymbol{z}$ and use the above formula. We can also use a composition of functions $f(\boldsymbol{x}) = (f_1 \circ \cdots \circ f_k)(\boldsymbol{z})$ to define more complex distributions, using Eq. 4 at every step.

The main challenge of normalizing flow models is defining the function $f$. First, we want to have a way to efficiently calculate the inverse $f^{-1}$. Second, calculating the determinant of the Jacobian becomes prohibitively expensive as the dimension $d$ grows. We often use $f$ with a special Jacobian form, e.g. the determinant of a lower triangular matrix is simply the product of its diagonal entries.

In all of our models we parametrize $f^{-1} : B^n \to B^n$ that transforms the input set $X$ to a set $Z$, with its corresponding base density. To make sure $p(X)$ is symmetric we have to satisfy two conditions. The base density $q(Z)$ should be permutation invariant. The mapping $f^{-1}$ should be permutation equivariant, meaning any permutation of the input permutes the output in the same way (Papamakarios et al., 2019, Section 5.6, Lemma 1). We will see different ways to enforce this, along with how to handle varying input size $n$. A detailed implementation description is in Appendix B.1.

**Inhomogeneous Poisson process.** Given an observed set $X$ we would like to use Eq. 2 for maximum likelihood training. Since the model assumes the independence between points we can define a normalizing flow on $B$, transform each point independently and get $p(X) = \prod_i p(\boldsymbol{x}_i)$ with Eq. 4. Invariance is achieved trivially. In the experiments, the flow is parametrized with coupling layers (Dinh et al., 2017) that have tractable inverse and determinant. Inside of coupling layers, elementwise functions are used, in particular splines with $K$ knots (Durkan et al., 2019).

**Autoregressive model with canonical ordering.** The idea to order the points by some arbitrary dimension allows us to use Eq. 1 for training. The flow acts on densities over $B^n$. Since the

ordering of the elements is known, a common approach is to use an autoregressive transformations that mimic the conditional probability formula by conditioning each element on all the previous, $z_i = f_{\boldsymbol{\theta}}^{-1}(\boldsymbol{x}_i | \boldsymbol{x}_1, \ldots, \boldsymbol{x}_{i-1})$ (Kingma et al., 2016). The ordered sequence of points $(\boldsymbol{x}_1, \ldots, \boldsymbol{x}_n)$ is passed to $f_{\boldsymbol{\theta}}^{-1}$, parametrized with a recurrent neural network. This scales to different set sizes $n$. Stacking multiple such layers together with set coupling gives a model like in Bender et al. (2020).

**Variational autoencoder and exchangeability.** Another way to ensure exchangeability is to introduce a latent variable $z$ such that the points $\boldsymbol{x}_i$ are conditionally independent given $z$. Further, we want to maximize the log-likelihood $\log p(X) = \log \int p(X, z) dz$ (Eq. 3) (Yuan et al., 2020). However, the true posterior $p(z|X)$ is intractable so we approximate it with a variational distribution $q(z|X)$. We can now maximize the likelihood by maximizing the evidence lower bound:

$$\log p(X) \geq \mathbb{E}_q \left[ \log p(X|z) \right] - D_{\text{KL}} \left[ q(z|X) || p(z|X) \right]. \tag{5}$$

Using amortized inference, parameters of $q(z|X)$ are defined as a function of $X$, and taking gradients w.r.t. the samples is enabled with the reparametrization trick (Kingma & Welling, 2014). We use a standard posterior approximation — a factorized normal distribution $\mathcal{N}(\mu(X), \sigma(X))$. To ensure the functions $\mu$ and $\sigma$ are permutation invariant, we use deep sets (Zaheer et al., 2017), or set transformers (Lee et al., 2019). The last thing we still need to calculate is $\log p(X|z)$. Each point $\boldsymbol{x}_i \in X$ is conditionally independent given $z$. In practice, this means we have an inhomogeneous Poisson process, but conditioned on a latent variable, i.e. a normalizing flow whose parameters are a function of $z$. To get a new realization we first sample $z$, then sample $\boldsymbol{x}_i$ from a normalizing flow conditioned on $z$. To make a fair comparison with other methods we implement an importance weighted autoencoder with a tighter likelihood lower bound (Burda et al., 2015).

## 4 CONFET MODEL

The models from the previous section trade-off between flexibility, having no special assumptions on the data structure, and modeling exact likelihood. Here, we introduce our new model, a continuous normalizing flow with equivariant transformations (CONFET), that satisfies all of the prerequisites.

If we want to have an expressive generative model that can both sample and estimate likelihood, we want to use a normalizing flow. However, having interactions between points leads in most cases to intractable inverses and determinants. This is not the case when using continuous normalizing flows. A CNF transforms an initial sample $z \sim q(z)$ to a sample in the target distribution $p(\boldsymbol{x})$ with a differential equation $f(z(t), t) = \partial z(t) / \partial t$. The change in log-density is (Chen et al., 2018):

$$\frac{\partial \log p(z(t))}{\partial t} = -\text{Tr} \left( \frac{\partial f}{\partial z(t)} \right) \tag{6}$$

and the final log-density is obtained by integrating across time, e.g., using a black-box ODE solver:

$$\log p(\boldsymbol{x}) = \log p(z(t_1)) = \log p(z(t_0)) - \int_{t_0}^{t_1} \text{Tr} \left( \frac{\partial f}{\partial z(t)} \right) dt. \tag{7}$$

The inverse is obtained by integrating in the opposite direction, $t_1$ to $t_0$. The only constraint on $f$, besides permutation equivariance, is that it needs to be, along with its derivatives, Lipschitz continuous which is easily satisfied by using smooth activations in a neural network. The benefit of this approach is that we are restricting $f$ mildly, meaning the dimensions can interact with each other arbitrarily. We utilize this to model interactions between points in a point process.

**Equivariant transformation.** A general way to define an equivariant map $f : B^n \to B^n$ is to define an invariant function $g : B^n \to \mathbb{R}$, and let $f(X) = \nabla_X g(X)$ (Papamakarios et al., 2019). However, this is computationally costly and numerically unstable (Köhler et al., 2020). We can instead use an explicitly defined equivariant layer (Zaheer et al., 2017; Maron et al., 2020):

$$f(X)_i = g(\boldsymbol{x}_i) + \sum_{j \neq i} h(\boldsymbol{x}_j), \tag{8}$$

where $g, h : B \to B$ are linear functions. We stack multiple layers with activations in between.

An alternative equivariant mapping is defined with a (multihead) self-attention layer (Vaswani et al., 2017). Lee et al. (2019) propose an extension for large sets that uses a small number of learnable inducing points $m \ll n$ which reduces the computational cost. The detailed discussion is in B.2.

Figure 2: Illustration of different interactions (with colored weights) and the corresponding Jacobian decoupling for a function $f$ and input set elements $\boldsymbol{x}_1, \boldsymbol{x}_2 \in \mathbb{R}^3$.

**Computing the trace** in Eq. 6 during training is expensive since it scales quadratically with the number of points $\mathcal{O}(n^2 d^2)$. We propose three different solutions to this: 1) using a trace estimator, 2) having a closed-form expression, and 3) decoupling the network to cheaply compute the trace. In the following, we make a distinction between the models based on which of these methods they use.

CONFET-STOCHASTIC. We use Hutchinson's trace estimator (Hutchinson, 1989) during training as proposed by Grathwohl et al. (2019). This reduces the cost to $\mathcal{O}(nd)$. However, it introduces additional variance into training making it potentially unstable.

CONFET-FIXED. Another approach is having a closed-form result for the trace. Notice how Eq. 8 decouples the mapping into a pointwise transformation and an influence from other points. Then a simple function with a trace equal to 0 is $f(X)_i = \sum_{j \neq i} h(\boldsymbol{x}_j)$. This connects to the early models in traditional normalizing flows (Dinh et al., 2015). In this case, the CNF is a single flow layer and we add additional elementwise coupling layers to model the inhomogeneous part of the process.

CONFET-EXACT. We extend the decoupling idea further to model arbitrarily complex architectures with cheap exact trace calculation. Chen & Duvenaud (2019) show how a neural network can be viewed as a combination of per dimension mappings $x_i \mapsto y_i$ and interactions between the dimensions $x_j \mapsto y_i$. They separate them explicitly into a transformer $\tau$ and conditioner $c$, e.g. $y_i = \tau(x_i, c(x_j))$, without losing expressiveness. This factors the Jacobian into a diagonal $(\partial \tau_i / \partial x_i)$ and hollow matrix $(\partial \tau_i / \partial x_j)$, meaning we only need to calculate the diagonal to obtain the trace. This can be done cheaply with modern deep learning frameworks. For example, the interaction between the points in Eq. 8 detaches itself naturally from the computation graph and we decouple the pointwise transformation $g$ with a masked neural network (Germain et al., 2015). We take this idea further by exploiting it hierarchically (Figure 2): after the dimensionwise mapping, we first impose interactions within a point, subsequently interactions between points. To summarize, one dimension of one point $\boldsymbol{x}_i \in X$ is transformed conditioned on all the other points in the set and all the other dimensions in the point. This decoupling allows us to obtain the exact trace in $\mathcal{O}(nd)$. Implementation details for Eq. 8 and attention network are in Appendix B.4.

**Training.** We can either use a fixed-step or adaptive ODE solver for Eq. 7. The trade-off is between having a bounded number of solver steps and a bounded error. To regularize ODE dynamics we can apply weight decay on the parameters of $f$ or penalize the Jacobian Frobenius norm (Finlay et al., 2020). With this, we can often resort to fixed-step solvers that offer significant speed improvements. Instead of backpropagating through solver steps, we use a memory efficient adjoint method to obtain the gradients (Farrell et al., 2013; Chen et al., 2018).

## 5 RELATED WORK

**Point processes.** Cox process (Cox, 1955) is a doubly stochastic Poisson process that defines intensity as a realization of a random field, e.g. a Gaussian process. We can think of this as a two step procedure where we first sample the intensity function and then sample points based on this intensity. A Neyman-Scott process (Neyman & Scott, 1958) generates parents from a Poisson process, and children around parents. A more specific example is a Matérn cluster process (Matérn, 2013) that places children in balls centered around parents. A variation where children follow a normal distribution is called a modified Thomas process (Thomas, 1949). Explicit interactions can be defined with a Gibbs process by assigning energy values to point configurations, stemming from statistical physics (Ruelle, 1969). However, this comes at a cost of calculating the normalization constant to find densities. Besides using pseudolikelihood (Besag, 1975) to fit them, other works used logistic composite likelihood (Clyde & Strauss, 1991) and Monte Carlo methods (Huang & Ogata, 1999).

Even though we do not specify the random field explicitly, the randomness from the base density is allowing us to model these processes with equivariant transformations. We avoid calculating the normalization constant by using CNFs that give us likelihood directly at no flexibility trade-off.

**Continuous normalizing flows.** Neural ODEs were proposed recently (Lu et al., 2018; Chen et al., 2018) as a continuous equivalent of the widely adopted residual architecture (He et al., 2016). By taking advantage of its properties we can construct a continuous version of a normalizing flow. Grathwohl et al. (2019) demonstrate the benefits of this approach by using a stochastic trace estimator that allows them to scale to bigger datasets. Having free-form Jacobian proved to be useful especially in our use case where we require it from equivariant transformations (beyond invariance through independence). Many recent works focused on improving the neural ODEs, including better representation power (Dupont et al., 2019), improved training (Gholami et al., 2019; Zhuang et al., 2020), regularization of the solver dynamics (Finlay et al., 2020; Kelly et al., 2020) etc.

The idea of using CNFs together with equivariant functions has been proposed before. Köhler et al. (2020) use it in the context of multi-body physical systems by modeling the Gibbs distribution with a simple equivariant function, calculating the forces between points proportional to their distance. In particular, they use a transformation based on a Gaussian kernel which allows them to have a closed-form trace solution. In Section 4 we offer ways to use more expressive equivariant functions with closed-form trace that do not rely on a predefined kernel. We also show how this can be applied for modeling point processes with exact likelihood — an important, previously unsolved problem.

A concurrent work by Li et al. (2020) proposes modeling sets similar to our STOCHASTIC model. Our work takes a general approach with further variations, and is more efficient and scalable.

Toth et al. (2020) propose modeling physics systems by transforming the initial samples with a Hamiltonian dynamics. The method is similar to CNFs in general, but it preserves volume, like our CONFET-FIXED model. Training requires variational approximation.

**Modeling sets.** Zaheer et al. (2017); Qi et al. (2017) achieve permutation equivariance similar to Eq. 8. They both show the universal approximation property of such a network. Same property holds for the attention based network (Lee et al., 2019). Wagstaff et al. (2019) study the theoretical limitations of invariant functions with sum aggregation. Other aggregation schemes include max-pooling, Janossy pooling (Murphy et al., 2019), featurewise sort pooling (Zhang et al., 2020), constructing the adjacency matrix and using graph methods (Grover & Leskovec, 2016) etc. We can use any of these methods in our model as well.

Korshunova et al. (2018) present a model that transforms every point independently and has exchangeable Student-t distribution for base density. Bender et al. (2020) extend this by having an autoregressive transformation and train on ordered sequences using Eq. 1. Our autoregressive model is very similar to this formulation. Other autoregressive models for set-like data include graph generation on breath-first search ordering (You et al., 2018); and ordering on z-axis for meshes (Nash et al., 2020) and point clouds (Sun et al., 2020). Neural models for temporal point processes know the explicit order and usually use RNNs to define density (Du et al., 2016; Mei & Eisner, 2017).

In contrast to autoregressive methods, Yang et al. (2019) model point clouds in a similar way to our variational autoencoder, with a CNF for posterior and a shared CNF for points. Yuan et al. (2020) model spatial point processes with a variational autoencoder. Instead of intensity, they also use density to avoid calculating the integral in Eq. 2 and define it with a nonparametric kernel. We generalize this by using normalizing flows as expressive density models. Finally, they demonstrate an application of spatial point processes to recommender systems by embedding the non-spatial data with a graph neural network, which is something our model can support as well.

## 6 EXPERIMENTS

In the following section we show how our model compares to competitors presented in Section 3 on various synthetic datasets that are both inhomogeneous and with clustering behavior. We also use real-world datasets that provide realistic location modeling scenarios and are commonly used in the literature. We also demonstrate the sampling capabilities of our models and compare their scalability and efficiency directly.

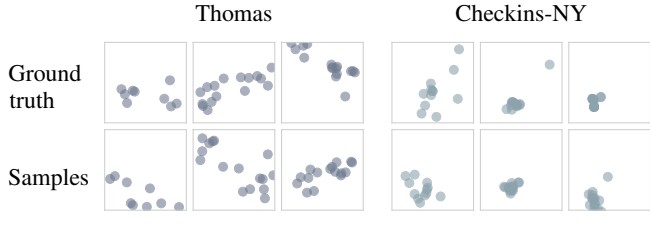

Figure 3: CONFET model has good sampling quality.

Figure 4: Out-of-dist. detection.

| | Autoregressive | IHP | IWAE | CONFET-FIXED | -EXACT | -STOCHASTIC |
|---|---|---|---|---|---|---|
| Check-ins NY | -1.69 ± 0.02 | -1.61 ± 0.02 | -2.03 ± 0.07 | -1.79 ± 0.05 | **-2.30 ± 0.09** | -1.85 ± 0.02 |
| Check-ins Paris | -3.39 ± 0.05 | -2.87 ± 0.04 | -3.57 ± 0.02 | -3.42 ± 0.03 | -3.43 ± 0.02 | **-3.58 ± 0.03** |
| Crimes | -1.61 ± 0.06 | -2.34 ± 0.01 | **-2.35 ± 0.00** | -2.34 ± 0.01 | -2.34 ± 0.01 | -2.23 ± 0.02 |
| Matérn | -0.38 ± 0.02 | -0.21 ± 0.00 | -0.59 ± 0.01 | -0.45 ± 0.00 | -0.97 ± 0.01 | **-1.03 ± 0.01** |
| Mixture | -1.55 ± 0.01 | -2.06 ± 0.00 | -2.05 ± 0.01 | -2.04 ± 0.01 | **-2.07 ± 0.00** | -2.02 ± 0.03 |
| Thomas | 0.13 ± 0.02 | -0.01 ± 0.00 | -0.23 ± 0.00 | -0.22 ± 0.00 | -0.36 ± 0.01 | **-0.55 ± 0.00** |
| Uniform | -0.41 ± 0.01 | 0.00 ± 0.00 | -0.48 ± 0.01 | -0.07 ± 0.00 | -0.49 ± 0.01 | **-0.51 ± 0.03** |

Table 1: Test loss mean and standard deviation for different models and datasets. Lower is better.

## 6.1 DATASETS

**Synthetic data.** For all datasets we simulate 1000 realizations on $(0, 1) \times (0, 1)$ area. Thomas dataset is simulated by first sampling $m \sim \text{Pois}(3)$ parents uniformly, and then for each of them we sample $n_i \sim \text{Pois}(5)$ points from normal distribution centered around the parents, with diagonal covariance with value $0.01$. Matérn generates the parents the same way, but children are uniformly sampled on a circle with $R = 0.1$. Mixture is an inhomogeneous process that generates $n \sim \text{Pois}(64)$ points from a mixture of 3 normal distributions with means $(0.3, 0.3)$, $(0.5, 0.7)$, $(0.7, 0.3)$ and diagonal covariance $0.05$. Uniform is a mixture of two uniform processes, as described in Section 1.

**Real-world data.** Check-ins NY[1] is a dataset of locations collected from social network users. We took all the data points from an area in New York and users that have more than 10 and less than 100 locations giving us 1938 unique users. We consider a set of all locations for a user as a single realization. We also construct a smaller dataset for a different city (Check-ins Paris) that has 286 unique users. Crimes[2] dataset contains daily records of locations and types of crimes that occurred in Portland. Each day contains between 298 and 736 points with 480 on average.

## 6.2 MODELS

Inhomogeneous Poisson process (IHP) is implemented as a density estimator $p(X) = \prod_i p(x_i)$ since it assumes independence between the points. We use a normalizing flow with spline coupling layers. Autoregressive model inputs points sorted on the first dimension, with Eq. 1 used for training. We stack autoregressive and set coupling layers (Bender et al., 2020). Importance weighted autoencoder (IWAE), as described in Section 3 uses either deep sets or attention layers as an invariant function that outputs posterior parameters. In both we use max-pooling for aggregation. Between attention layers we apply layer normalization (Ba et al., 2016). Term $p(X|z)$ is an IHP conditioned on $z$. We use the same hyperparameter ranges. During training we use 5 samples, to estimate the test likelihood (Eq. 3) we use 5000 samples (Burda et al., 2015). Models from Section 4 are implemented as a CNF with equivariant transformations that are either deep set or attention layers. We usually use dopri5 solver (Dormand & Prince, 1980). CONFET-FIXED uses Runge-Kutta solver with 10 steps. Further details on hyperparameters and ablation studies are in Appendix D.

Datasets are split into training, validation and test sets (60%-20%-20%). We train with early stopping after validation loss does not improve for 20 consecutive epochs. We use mini-batches of size 64 and Adam optimizer with learning rate of $10^{-3}$ (Kingma & Ba, 2015).

---

[1](Cho et al., 2011) `https://snap.stanford.edu/data/loc-gowalla.html`
[2]`https://nij.ojp.gov/funding/real-time-crime-forecasting-challenge`

| | Autoregressive | IHP | IWAE | CONFET-FIXED | -EXACT | -STOCHASTIC |
|---|---|---|---|---|---|---|
| Check-ins NY | 0.0157 | 0.0354 | 0.0346 | 0.0299 | **0.0120** | 0.0232 |
| Check-ins Paris | 0.0224 | 0.0264 | 0.0331 | 0.0241 | **0.0144** | 0.0192 |
| Crimes | 0.0056 | 0.0024 | 0.0024 | 0.0024 | **0.0017** | 0.0044 |

Table 2: Wasserstein distance of the sample distribution and the ground truth. Lower is better.

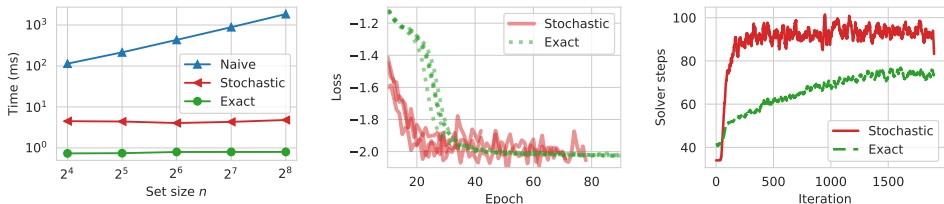

Figure 5: Exact trace calculation is faster (left), has more stable training (middle), and requires less function evaluations (right), compared to the stochastic trace estimation.

## 6.3 RESULTS

The loss we use is per-point negative log-likelihood, i.e. $\mathcal{L}(X) = -\log p(X)/n$. We report the mean and variance of the loss on a held-out test set averaged over 5 runs for models selected based on their validation set performance. The results are in Table 1. Our models achieve the best scores on most of the datasets. It is worth noting that CONFET-FIXED is often on par with others even though it has the simplest architecture. Figure 4 shows how we can use our model trained on Checkins-NY dataset to detect the out-of-distribution samples, in this case from Checkins-Paris. We also note our model outperforms (Köhler et al., 2020) by a large margin (see Appendix C.2).

**Sampling quality.** We provide qualitative samples for CONFET-EXACT model on Thomas and Checkins-NY dataset in Figure 3. We show it is able to learn the clustering process, and in the second example additionally learns the underlying city topology that restricts the samples, e.g. to appear only on the peninsula and not in the water. Additional samples for all the models are in Appendix C.3. We also assess the quantitative quality of the samples on real-world datasets by comparing the distributions of in-between point distances in real data and model samples. The results in Table 2 suggest our models produce samples of higher quality.

**CONFET model comparison.** In Section 4 we provided three different CONFET models based on how they calculate the trace. We have seen that CONFET-EXACT and CONFET-STOCHASTIC perform the best. Now we compare them directly. Figure 5 contains the results. First, we test the speed of trace calculation by timing it with different set sizes. We use an identical architecture for both models, further details are in Appendix C.4. We can see that CONFET-EXACT is faster although both have the same theoretical cost $\mathcal{O}(nd)$. The naive exact trace calculation proves to be unusable in practice. Second, we test the training stability on Mixture dataset and notice that both converge to the same value but CONFET-STOCHASTIC has very noisy loss curve. Finally, we plot the number of function evaluations of the dopri5 solver trained on Matérn dataset during training. Here again, CONFET-EXACT achieves the better result.

We conclude that CONFET can be used in most cases to model sets and point processes. If we want better scalability while retaining the same performance we should use CONFET-EXACT. In case we model simple interactions or inhomogeneous data CONFET-FIXED will work well.

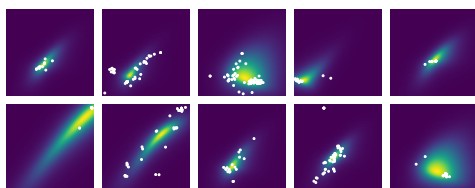

Figure 6: Density realizations for NY data.

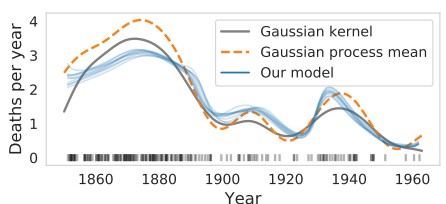

Figure 7: Comparison to GP-PP on coal dataset.

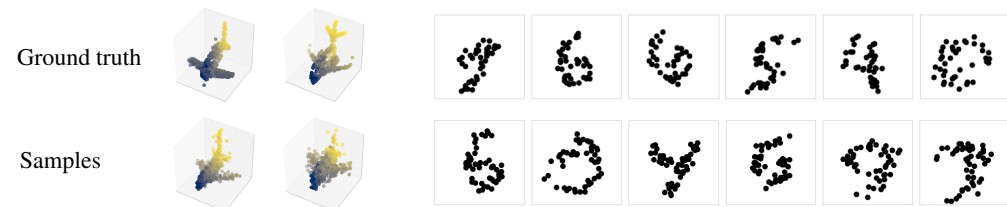

Figure 8: Samples of airplane point clouds (left) and discretized digits (right).

### 6.4 STOCHASTIC INTENSITY DIRECTLY FROM CONFET MODEL

Since the density $p(X)$ in CONFET model is influenced by the interactions between the points, it will not be stationary but will depend on the locations in $X$ (unlike e.g., IHP model). Therefore, we can draw one density realization on the region $B$ by sampling $X \sim p(X)$. The density can also be easily evaluated at non-observed points $x \notin X$. Figure 6 shows different density realizations and the corresponding samples from a model trained on Check-ins NY data. This connects to the doubly stochastic processes, in particular Cox process (Cox, 1955).

A popular way to define a Cox process is by defining the random field with a Gaussian process (Rasmussen & Williams, 2006), and drawing an intensity function as a random realization. To ensure the intensity is always non-negative, the random function $f \sim \mathcal{GP}$ can be squared (Lloyd et al., 2015) or we can apply a sigmoid function (Adams et al., 2009). Figure 7 shows how a Gaussian process modulated poisson process (GP-PP) (Lloyd et al., 2015) fits the coal mining disaster dataset with 191 points collected from 1851 to 1962. We additionally plot the empirical intensity measure estimated with a Gaussian kernel and notice that GP-PP overestimates the point counts on some parts of the data. Finally, we plot 10 intensity function realizations from our model. CONFET closely matches the underlying intensity and shows uncertainty in parts where it observed less data.

### 6.5 MODELING POINT CLOUDS AND DIGITS

To further test the capabilities of our model we use two traditional set modeling problems: point clouds and discretized images. We use airplane samples from a point cloud dataset (Wu et al., 2015), and uniformly sample 512 spatial points to reduce the input size. We stack 8 continuous normalizing flow transformations and train using maximum likelihood. For digits dataset (LeCun et al., 1998) we randomly sample 50 white pixels in each image and record their 2-D positions. In Figure 8 we see samples from our model compared to the real data. We conclude our model is able to capture the shapes which shows that it can be used for complex set data.

## 7 CONCLUSION AND FUTURE WORK

In this work we presented a novel model for point processes and random sets that uses normalizing flows with equivariant functions. Throughout the experiments we showed that it is able to model complex processes with interactions between the points while having tractable likelihood and straightforward sampling. The performance of our model is better than all the competitors. We additionally provide different versions that are more scalable while retaining the same properties of the base model. Our model relies in particular on two types of deep learning models: (continuous) normalizing flows and equivariant neural networks. By utilizing new techniques developed in these two subfields we hope our method will perform even better in the future.

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

## A    THEORETICAL BACKGROUND

### A.1    CARDINALITY IN LIKELIHOOD

In this section we discuss how to define the likelihood for point processes. Let's take an inhomogeneous Poisson process as an example. It defines the density on $B$ and the probability of observing a set $X$ is $p(X) = p(\boldsymbol{x}_1, \dots, \boldsymbol{x}_n)p(n)$. In our experiments we considered only the first part, ignoring the cardinality of the set. This leads to some unexpected behavior. For example, we will almost always get higher probability for sets with a smaller cardinality because, e.g., $p > p^2$, when $p < 1$, and vice versa. This is also pointed out by Vo et al. (2018). They propose the following likelihood:

$$p(X) = p(n)n!U^n p(\boldsymbol{x}_1, \dots, \boldsymbol{x}_n), \tag{9}$$

where $U$ is the unit of hyper-volume to ensure consistent results over different measurement units. However, our approach is not completely wrong for the following reasons. First, we consistently use the same metric across all the datasets and models. As we have shown in Section 2, a point process can be deconstructed into modeling cardinality $p(n)$ and modeling point locations. Thus, all of the models can be trivially extended to include $p(n)$ and the relative differences in the likelihood would stay the same since the additional term would be a constant. Second, we use per-point likelihood which tells us how much mass is allocated on each point on average. This, however, ignores cardinality in a different way. Finally, we use this to directly compare to other related work done on modeling sets. To summarize, we can compare models with only $p(\boldsymbol{x}_1, \dots, \boldsymbol{x}_n)$ but need to include $p(n)$ when comparing instances inside the dataset.

Figure 9 illustrates the difference perfectly. Having a ground truth model $\mathrm{Pois}(64)$ of on inhomogeneous Poisson process means that the sample with $n = 64$ elements should have higher probability than $n = 70$. But since the per-point likelihood is larger than 1, this is not the case. By including the $p(n)$ term, we flip the difference and the probability is now correctly assigned. When using just per-point likelihood, the cardinality is completely ignored and sets have the same probability.

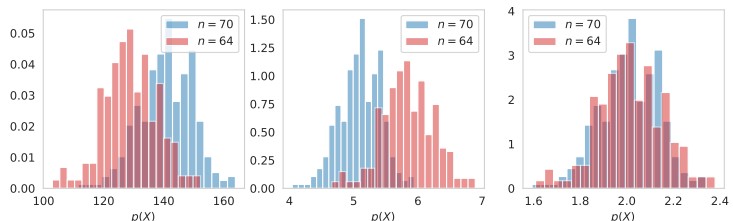

Figure 9: Distribution of probabilities of 300 samples with $n = 70$ and $n = 64$ evaluated on model with ground truth $\mathrm{Pois}(64)$. Left to right: $p(\boldsymbol{x}_1, \dots, \boldsymbol{x}_n)$; $p(\boldsymbol{x}_1, \dots, \boldsymbol{x}_n)p(n)$; and $p(\boldsymbol{x}_1, \dots, \boldsymbol{x}_n)/n$.

### A.2    EQUIVARIANT DRIFT GIVES INVARIANT DENSITY

**Theorem 1** (Equivariant flows). *(Adapted from (Papamakarios et al., 2019, Section 5.6, Lemma 1)) Let $p(X)$ be the density function of a flow-based model with transformation $f : \mathbb{R}^{n \times d} \to \mathbb{R}^{n \times d}$ and base density $q(Z)$. If $f$ is equivariant with respect to $\Gamma$ and $q$ is invariant with respect to $\Gamma$, then $p(X)$ is invariant with respect to $\Gamma$.*

*Proof.* See (Papamakarios et al., 2019). □

Theorem 1 is satisfied for permutation group $\Gamma$ if we, e.g., use normal or uniform base distribution, and an equivariant transformation.

**Theorem 2** (Equivariant ordinary differential equation). *Let $f(X)$, $f : \mathbb{R}^{n \times d} \to \mathbb{R}^{n \times d}$ be the solution of the ordinary differential equation $dZ(t)/dt = g(Z(t), t)$ on $I = [t_0, t_1]$ with an initial condition $Z(t_0) = X$. If $g$ is equivariant with respect to $\Gamma$ then so is $f$.*

*Proof.* $\gamma f(X) = \gamma f(Z(t_0)) = \gamma Z(t_0) + \gamma \int_{t_0}^{t_1} g(Z(t), t)dt = \gamma Z(t_0) + \int_{t_0}^{t_1} g(\gamma Z(t), t)dt = f(\gamma Z(t_0)) = f(\gamma X), \gamma \in \Gamma.$ □

# B  IMPLEMENTATION

In the following we describe in detail the implementation of the normalizing flows, in particular the traditional ones for baseline models. We later describe attention, self-attention and induced attention for sets. We provide the background and implementation of efficient trace calculation for equivariant transformations. We additionally discuss mini-batch training and CONFET architecture.

## B.1  NORMALIZING FLOW LAYERS

In the following we review the traditional normalizing flows, i.e. different ways to define $f : \mathbb{R}^d \to \mathbb{R}^d$ with the tractable inverse and the determinant of the Jacobian. We denote random variables $z, x \in \mathbb{R}^d$ coming from the base and target density $q$ and $p$, respectively. The forward map $z \mapsto x$ is parametrized with $f$, and the inverse map $x \mapsto z$ with $f^{-1}$. In all of the models, the analytical inverse exists. In practice we often focus on parametrizing the inverse map, allowing density estimation and maximum likelihood training.

**Coupling layer** (Dinh et al., 2015; 2017) defines the transformation of one part of the data point $x$ conditioned on the rest. If $x \in \mathbb{R}^d$, we take first $k$ dimensions and copy them, $x_{1:k} = z_{1:k}$. Rest go through an affine transformation with coefficients as a function of $z_{1:k}$:

$$x_{k+1:d} = x_{k+1:d} \odot \exp(s(z_{1:k})) + t(z_{1:k}), \tag{10}$$

where $s, t : \mathbb{R}^k \to \mathbb{R}^{d-k}$ are unrestricted neural networks. Notice the Jacobian has a special form where the only non-zero elements are on the diagonal and in one block under the diagonal. The determinant is then the product of the diagonal entries. Inverse is obtained by noticing $z_{1:k} = x_{1:k}$, and then $z_{k+1:d} = (z_{k+1:d} - t(x_{1:k})) \odot \exp(-s(x_{1:k}))$. The computation complexity is the same in both the sampling and density estimation directions.

The function described in Eq. 10 is called an affine transformation. Instead, we can use more expressive functions such as rational-quadratic splines. Details of implementation can be found in (Durkan et al., 2019)[3]. It defines a monotonic function with $K$ bins where each bin is a rational-quadratic function. Increasing $K$ increases the expressiveness. The inverse and determinant are easy to obtain. In the context of coupling layers, one part of the data defines the spline parameters that transform the rest.

**Autoregressive layer** transforms the $i$th dimension based on all the previous values $x_{:i}$. We can implement this by processing a sequence $(z_1, \ldots, z_d)$ with a neural network, in our case with a recurrent neural network that outputs the parameters of an affine transformation similar to the coupling layer. The Jacobian is now a lower triangular matrix. When calculating the forward direction, we know all the values of $z$, and can use efficient parallel implementations making the calculation fast. However, when inverting element $x_i$ we need to know all of the previously inverted values $z_{:i}$. This means that autoregressive layer is inherently slow and sequential in one direction. Therefore, parametrizing $x \mapsto z$ direction is preferred for maximum likelihood training (Papamakarios et al., 2017), and for fast sampling, we would parametrize the $z \mapsto x$ direction (Kingma et al., 2016).

**Combining layers** is implemented as a composition of functions $f_1 \circ \cdots \circ f_k$. In practice, for each transformation we know $f$ and $f^{-1}$ together with the log-determinant of the Jacobian. Therefore, for input $z_0$ we calculate $z_1 = f_1(z_0), \ldots, z_k = f_k(z_{k-1})$. We accumulate the log-determinant at every step and calculate $p(z_k)$ using Eq. 4. The same principle applies for the inverse direction.

**Why don't we use CNFs in all models?** We do not use the continuous normalizing flows in models from Section 3 for two reasons. First, some models represent existing works in literature that we reimplement here, e.g. (Bender et al., 2020). Second, CNFs are not necessarily always better in terms of efficiency and final performance given the task. Both the variational autoencoder and inhomogeneous process are defining the density on $B$ without any constraints. Since invariance is implemented via independence, CNFs are here as good as any other normalizing flow method. On the other hand, if we actually want to model interactions, like we do with CONFET, we need to have a model that supports full Jacobian and CNFs are a perfect candidate.

---

[3]https://github.com/bayesiains/nsf

### B.2 SELF-ATTENTION

Attention originated in natural language processing (Vaswani et al., 2017). Formally, given matrices $Q \in \mathbb{R}^{n_q \times d_k}, K \in \mathbb{R}^{n_v \times d_k}, V \in \mathbb{R}^{n_v \times d_v}$ of queries, keys and values, respectively, the attention layer is defined as:

$$\text{Attention}(Q, K, V) = \text{softmax}\left(\frac{QK^T}{\sqrt{d_k}}\right) V.$$

Vaswani et al. (2017) introduce the multihead attention, an extension where we apply $h$ attention layers in parallel, with learnable parameters that transform each matrix with a linear layer. In our case, we are interested in a (multihead) self-attention layer that models explicit interactions between elements of sets:

$$\text{SelfAttention}(X) = \text{Attention}(X, X, X),$$

$X \in \mathbb{R}^{n \times d}$. For large sets, the interaction matrix will be of size $n \times n$. To reduce the computation cost, Lee et al. (2019) introduce the induced attention layer:

$$\text{InducedAttention}_m(X) = \text{Attention}(X, H, H)$$
$$H = \text{Attention}(I, X, X),$$

where $I \in \mathbb{R}^{m \times d}$ is a set of learnable $m$ points, $m \ll n$.

### B.3 ATTENTION NETWORK

The attention network is defined as follows. The input data of shape $(n, d)$ is projected with a linear layer to $(n, h)$. We then apply self-attention layers $H_1 = H_0 + \text{Attention}(H_0, H_0, H_0)$, where $H_0 = X$ (we can also use $\text{InducedAttention}_m$). We apply optional layer normalization, and a single linear layer mapping proceeded with an optional residual layer. After $L$ such layers we obtain final transformed set $H_L$, and project it back to $(n, d)$ with a linear layer. Using initial input and a single hidden layer neural network with sigmoid activation we obtain the gating set $G$. The final output of attention network is $Y = G \cdot X + (1 - G) \cdot H_L$, a weighted combination of the initial input and the attention output.

### B.4 EFFICIENT TRACE ESTIMATION

In this we discuss how to efficiently calculate the trace, without using stochastic methods. We first review the results from Chen & Duvenaud (2019). In general, if we have a neural network $f : \mathbb{R}^d \to \mathbb{R}^d$ we want to get the diagonal values of its Jacobian $\partial f(\boldsymbol{x})_i / \partial x_i$. A straightforward but expensive solution is to use built-in functionality of modern deep learning frameworks to calculate this sequentially for each $i$. The cost is then $\mathcal{O}(d^2)$.

Another way is to define $f(\boldsymbol{x})_i = \tau(x_i, \boldsymbol{h})$, where $h$ is a $d_h$-dimensional vector containing information about all the other values of input vector $\boldsymbol{x}$, i.e. a conditioner network $c$ that takes all $x_j, j \neq i$ as an input. Function $\tau$ maps $\mathbb{R}^{d_h+1} \to \mathbb{R}$. Now when we take derivative $\partial f(\boldsymbol{x})_i / \partial x_i$ it breaks into $\partial \tau_i / x_i$ and zero term from the conditioner. Therefore, we can detach the network $c$ completely from the computation graph and just calculate $\partial \tau_i / x_i$ for all $i$ in parallel. When backpropagating we need to attach $c$ back to allow learning of the parameters. More details can be found in Chen & Duvenaud (2019). If $d_h = d - 1$ we recover the original network. We often want to use much smaller values for efficiency.

In our case, we use sets of points. First step is to build an inhomogeneous transformation — a neural network that acts on individual points $\boldsymbol{x}_i \in \mathbb{R}^d$. This is the use case from above. We build a conditioner as a masked neural network (Germain et al., 2015) that gives a hidden vector $\boldsymbol{h} \in \mathbb{R}^{d_h}$ for each dimension in $\boldsymbol{x}_i$. We input the concatenation of $\boldsymbol{h}$ and a single dimension of $\boldsymbol{x}_i$ to a neural network with output dimension of 1. The trace is calculated as above. Next, we discuss how to extend this with interactions between the points.

**Sum-pooling** in Eq. 8 is defined as a sum of all the contributions from other points in the set. We apply a function $h : \mathbb{R}^d \to \mathbb{R}^d$ to each point $\boldsymbol{x}_i$ to get $\boldsymbol{h}_i$. Now the contribution for element $i$ can be simply written as $\sum_{j \neq i} \boldsymbol{h}_j = \sum_j \boldsymbol{h}_j - \boldsymbol{h}_i$. This way, we calculate $\sum_j \boldsymbol{h}_j$ only once. We can

extend this trivially to mean-pooling by dividing everything with $n - 1$. For max-pooling we would need to get the highest and second highest values of dimensions in all $\boldsymbol{h}_i$ and assign the highest to $i$ if its value is smaller, otherwise assign the second highest value. This way we simulate the max operation on the set excluding $i$th element.

**Attention** cannot be parallelized as easily. The issue is that a row in the similarity matrix is built by comparing current element with all the others. It cannot be decoupled and masking out the diagonal is not enough. A pragmatic approach is to first obtain the conditioner vectors with a masked neural network. Now, we can input them to any downstream network, including self-attention, without losing the decoupling property. Another approach is to calculate attention with all the elements excluding $i$, and use max-pooling to get the embedding $\boldsymbol{h}_i$. This is more expensive so we pursue the first option.

### B.5 TRAINING IN BATCHES

All of the described functions work with inputs of shape $(n, d)$. We train in mini-batches, i.e. process multiple sets at a time. Since $n$ varies between them, we pad them all to the biggest set with zeros and keep track of the original size. This is important when calculating the loss since we use per-point negative log-likelihood. Also, when implementing interactions between the elements, we want to remove those that represent the masking. For example, in the attention network, the elements interact with each other through the similarity matrix. If most of our inputs are zero padding, this will give unexpected results after calculating $\mathrm{softmax}$. Therefore, we zero-out the elements corresponding to the padding with infinity-masking. Same is done for other methods.

In case we forget to mask-out the padding values before aggregation, the model will learn slower, get worse results, and in extreme cases not train at all.

### B.6 CONFET IMPLEMENTATION

We assume data comes from a region $B$. For $\boldsymbol{x} \in B$, it holds that $x_i \in \boldsymbol{x}, x_i \in (0, 1), i = 1, \ldots, d$. This means all of the points fall on the unit square. We can achieve this by appropriately preprocessing the data. The introduced restriction is here to simplify the notation, without losing generality. The base density for all of the models is then form on $B$.

Since CNFs work on a real number domain we project the input data to $\mathbb{R}^{n \times d}$ with a logit transformation, apply CNF, and project it back with sigmoid transformation. The model is defined as a normalizing flow with three transformation, one of them being a CNF. Logit is an inverse of sigmoid and their Jacobians can be calculated in closed-form.

We integrate an ODE from $t_0 = 0$ to $t_1 = 1$. Although tuning these hyperparameters, or learning them during training is possible, we avoid doing that. If we change $t_1$ from 1 to 0.1, the network can adapt the weights to get the same output.

We extend the library provided by Chen et al. (2018)[4] to allow permutation invariant densities.

## C EXPERIMENT DETAILS

### C.1 DATASET PREPROCESSING AND GENERATION

For synthetic datasets we take care of the edge effects by simulating on a region larger than $(0, 1) \times (0, 1)$, e.g. for Matérn clustering process with $R = 0.1$ we extend the simulation square by $0.1$ on all sides. The points that fall out of the observed region are discarded. Thomas and Matérn are simulated as it is described in the main text.

Checkins-NY and Checkins-Paris use data of user location logs that are recorded over the period of 2009 and 2010. All locations for a user are a single realization of the underlying process. Initial data contains just under 6.5 million locations. We take the locations from two cities, New York, between coordinates $(40.6829, -74.0479)$ and $(40.8820, -73.9067)$, and Paris, between $(48.5, 2)$ and $(49, 3)$, respectively. Data contains very large sets, largest one having 904 elements.

---

[4] `https://github.com/rtqichen/torchdiffeq`

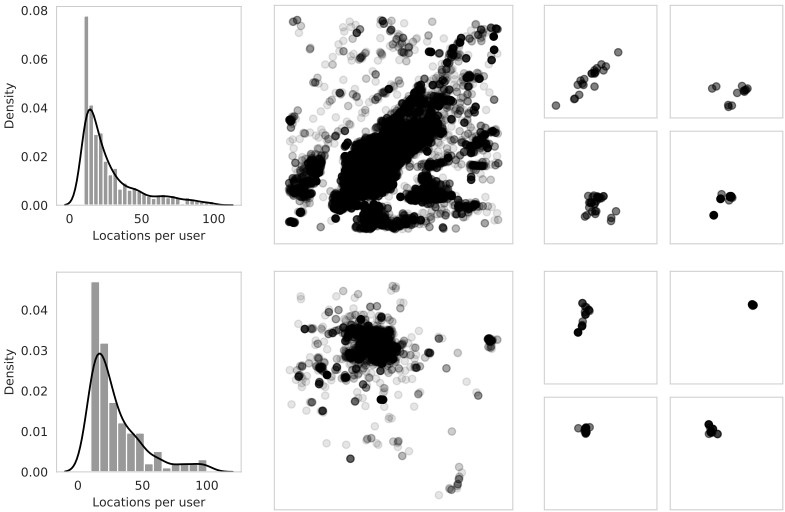

Figure 10: Upper row: Checkins-NY. Bottom: Checkins-Paris. From left to right: distribution of the number of locations per user after preprocessing; all the locations for all the users; sets of locations for four randomly sampled users.

|  | Mixture | Thomas | Checkins-NY |
|---|---|---|---|
| Original model | -1.0511 | -0.0134 | -0.6368 |
| With added inhomogeneous layers | -1.5583 | -0.0148 | -0.7244 |

Table 3: (Köhler et al., 2020) results on the datasets from Section 6.

Crimes dataset contains recordings of crimes that happened in the city of Portland from March 01 2012 to December 31 2012, with 309 days in total. It contains 146927 logs, each with a time, coordinate and type of case. We aggregate all the occurrences in one day to get a single realization. This gives us very large sets, smallest one having 298 and largest 736 elements.

## C.2 COMPARISON TO OTHER EQUIVARIANT FLOWS

In addition to the main experiments in Section 6, here we compare to the model by (Köhler et al., 2020) that also uses an equivariant flow with closed-form trace. The results are shown in Table 3 Their model is designed for physics simulations so it has to include rotation and translation symmetry. They achieve this by transforming the points based on their distances using a simple Gaussian kernel. Because of this limitation, it is unable to model the inhomogeneous part of the process as can be seen from the results. To combat this we added additional spline coupling layers but the final results are still much worse than our model. We conclude that our equivariant transformation (e.g. Eq. 8) is well suited for the problem of modeling sets and achieves better performance than all competing methods while having closed-form trace computation.

## C.3 ADDITIONAL SAMPLES

Figure 11 shows 6 samples drawn from models trained on Thomas dataset. In the first row we can see ground truth samples. The clustering behavior is not captured by IHP model. Other models are able to generate convincing samples.

We used the same random seed for sampling the initial set $Z \sim p(Z)$ in all CONFET models. Remarkably, the similarity of the output is prominent even though the architectures and parameters are different. We conclude that CONFET models learn similar distributions $p(X)$ given the same dataset.

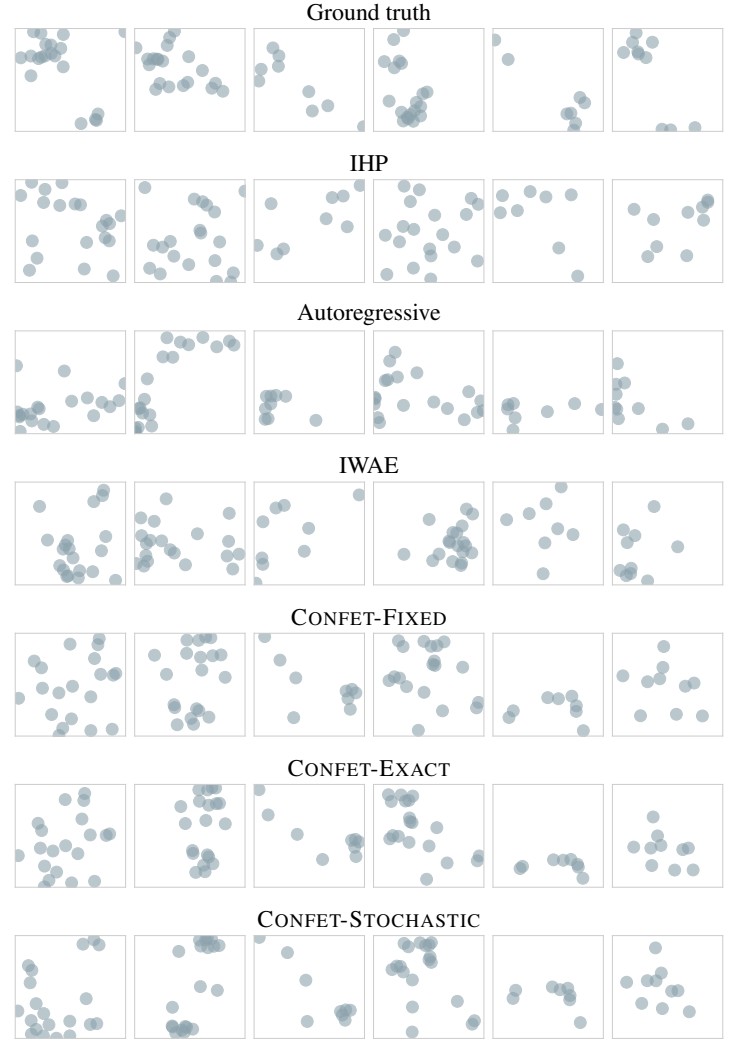

Figure 11: Comparison of samples from different models on Thomas dataset. Number of points is the same across columns. We use the same random seed for different CONFET models.

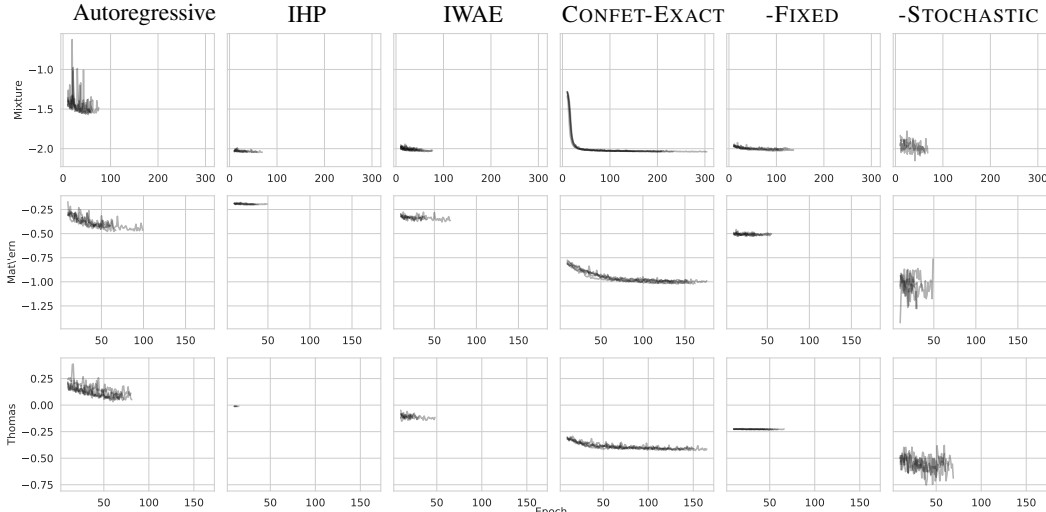

Figure 12: Training dynamic for baseline and CONFET models on Mixture, Matérn and Thomas.

|  | IHP | CONFET-FIXED (0) | CONFET-FIXED (5) |
|---|---|---|---|
| Mixture | **-2.06 $\pm$ 0.00** | -0.02 $\pm$ 0.00 | -2.04 $\pm$ 0.01 |
| Matérn | -0.21 $\pm$ 0.00 | -0.11 $\pm$ 0.00 | **-0.45 $\pm$ 0.00** |
| Thomas | -0.01 $\pm$ 0.00 | -0.15 $\pm$ 0.00 | **-0.22 $\pm$ 0.00** |

Table 4: Comparison of using CONFET-FIXED with and without additional coupling layers.

### C.4 SPEED COMPARISON

In Figure 5 (Left) we compare the time to calculate the trace using three different methods: naive calculation, one step of stochastic estimator and using a cheap exact computation as described before. We use the same CONFET-EXACT architecture for all three with input dimension of 2, hidden dimension 64, and a single attention layer. We vary the number of input set elements from $2^4$ to $2^8$. The experiment was repeated 5 times on the hardware described in C.6.

### C.5 TRAINING CURVES

In Figure 5 (Middle) we overlay the validation losses recorded during training for CONFET-EXACT and CONFET-STOCHASTIC. We picked two models that have the best final validation loss after hyperparameter search. We plot all 5 runs.

Figure 12 shows the validation loss change across training epochs for all our models on Mixture, Matérn and Thomas dataset. The chosen models are those that achieved the best average validation loss. The different number of epochs in different plots comes from the early stopping. We can notice that CONFET-STOCHASTIC has more noisy training compared to CONFET-EXACT and CONFET-FIXED because of its stochastic trace estimator.

### C.6 HARDWARE

We train all of our models on a single Nvidia GTX1080Ti GPU (12GB). The machine has 256GB RAM and an Intel Xeon E5-2630 v4 @ 2.20 GHz CPU.

## D ABLATION STUDIES

### D.1 HYPERPARAMETER SEARCH

In all of the models we tried placing the weight decay of $10^{-3}$ on the weights. This sometimes leads to more stable training and better performance. Using smaller learning rate ($10^{-4}$) makes training slow without other benefits. We did not try learning rate schedulers.

In IHP we use a normalizing flow with $L \in \{1, 5\}$ spline coupling layers, each spline defined with $K \in \{5, 10\}$ knots. Autoregressive model stacks $L_a \in \{5, 10\}$ autoregressive and $L_c \in \{5, 10\}$ set coupling layers. When using attention in IWAE, besides the number of layers $L \in \{1, 5\}$, we define the number of heads $H \in \{1, 8\}$. In all of the above, having more layers (bigger model) does not improve the result.

Equivariant transformations are defined with $L \in \{2, 5\}$ deep set or attention layers. We use mean or max-pooling for aggregation. We noticed that having 2 or 5 attention layers in CONFET-STOCHASTIC does not influence the results much. Because of that, using a bigger model is unnecessary for our datasets. We tried Frobenius norm regularization of an ODE (Finlay et al., 2020), but did not observe big changes in performance.

### D.2 THE ROLE OF ADDITIONAL LAYERS IN CONFET-FIXED

The CONFET-FIXED model has only interactions between the points in its drift function so it is unable to model the inhomogeneous part of the process. That is the reason why we add additional coupling layers, same as those in IHP, along with the CNF layer. Here, we compare the two versions of the model, one with 5 of these layers and other without any. We additionally include IHP into

|         | Induced Attention | Self Attention |
|---------|-------------------|----------------|
| Matérn  | -0.99 ± 0.00      | **-1.03 ± 0.01** |
| Mixture | **-2.02 ± 0.03**  | -1.92 ± 0.06   |
| Thomas  | -0.29 ± 0.09      | **-0.55 ± 0.00** |

Table 5: Comparison of using induced and regular self-attention in CONFET-STOCHASTIC.

comparison. The results in Table 4 show that we need the layers to achieve competitive results. They are on par with IHP on a synthetic inhomogeneous dataset. It performs much better on datasets with interactions. A version with no coupling layers has a better test loss score than IHP on Thomas dataset.

### D.3 WHEN TO USE INDUCED ATTENTION?

Table 5 shows the comparison between using induced ($m = 8$) and regular self-attention. We can see that induced attention usually performs worse. We did not tune the hyperparameter $m$ so the results can be different for the optimal value. Having an extra hyperparameter is a drawback, however, the lower memory footprint of induced attention can help train models on sets with very large cardinality.

