# OpenReview forum: "Equivariant Normalizing Flows for Point Processes and Sets"
_ICLR.cc/2021/Conference — Reject_

### Official Review · AnonReviewer4 · 2020-10-27
**Good contribution, well written; accept**

**Rating:** 8
**Confidence:** 3

**Review:**

*Summary
This paper provides a novel method of learning density models for sets of points by modelling the sets as samples from a point process, approximated with normalizing flows.  A point process gives a probability to a set of points, not assuming that the points are independent of one another. The authors describe the CONFET method, which uses continuous normalizing flows to map from a uniform point process with a learned transformation. The authors describe a method to tractably compute the exact trace of their transformation, allowing it to scale to high dimensions and numbers of points. Experiments show state-of-the-art performance on benchmarks.


*Positives
The paper is generally well-written and motivated well.
The exact computation of the trace in this setting is a key aspect to the tractability of this work, and is a good contribution.
The experimental results seem convincing that the overall method is an advance in the state-of-the-art of modelling point datasets.
The appendices are discursive and comprehensive

*Questions
How does this method perform on tasks such as modelling point clouds?

*Recommendation
Overall I recommend to strongly accept this paper.  The trace-computation method is a nice fundamental contribution, which is used to present a compelling set of experimental results.


*Minor points
Typo in paragraph 1 of section 6.2: 'studies are in D' should be 'appendix D'
Typo in paragraph 2 of section 2, 'Symmetry requirement comes from' should be 'the symmetry requirement...'
Figure 5 might be better off in a different colour scheme than green/red for those with colourblindness

---

> ### Author Response · Authors · 2020-11-19
> **Response to Review 4**
>
> We thank the reviewer for positive feedback.
>
> We updated the paper to include your points. We will update the color scheme in Figure 5 for the camera-ready version. We also included additional experiments including point clouds and discretized digits in Section 6.5.

---

> > ### Comment · AnonReviewer4 · 2020-11-23
> > **Reply**
> >
> > Thank you for your response and update to the paper.
> > After reading the paper again, all of the other reviewers' comments, and the additional experiments provided, I'll keep the score as it is. I understand that the reviewers have concerns about the novelty of the paper. At its core, the paper provides a novel (more efficient) method of computing the Jacobian of a model and experimental results showing this is superior to previous methods that provide a density. There are many papers published which provide fundamentally the same two points, such as [Finlay et al 2020] and [Kingma et al 2018], and so I don't agree this paper lacks enough novelty to be published.
> >
> >
> >
> > Chris Finlay et al., How to train your neural ODE: the world of Jacobian and kinetic regularization,  ICML 2020
> > Diederik P. Kingma et al., Glow: Generative Flowwith Invertible 1×1 Convolutions.

---

### Official Review · AnonReviewer1 · 2020-10-28

**Rating:** 5
**Confidence:** 4

**Review:**

Summary:
This paper proposes a method based on continuous normalizing flows that can model random sets of exchangeable points. The advantages of the method are as follows: (1) it can handle a complex density function of sets and (2) it is designed so that model learning can be performed based on tractable likelihood while considering dependencies between samples. The effectiveness of the proposed method is shown in the experiments using synthetic and real-world datasets.

Pros:
1. This paper is well-organized and its presentation is good. Related works are sufficiently cited.
2. It is a first attempt to unify point process models with normalizing flows, which might lead to a new potential research direction.
3. The authors present a new model, CONFET, which is well-designed so that the model is flexible and its learning is efficient.

Cons:
1. This work is a good combination of several recently-developed techniques and it does not include a great technical contribution.
2. The goal of this work is misleading to me. The proposed model, CONFET,  is for the probability density estimation of sets and may not be for point process intensity estimation. See detailed comments.

Reasons for score:
An attempt to unify point process models with normalizing flows is interesting. Also, the proposed method, CONFET, is well-designed. Overall, the paper is well-written. But, several points are misleading to me; this is important in the determination of the placement of this work in the research field. Accordingly, for now, my opinion is that this paper is marginal.

Detailed comments:
This work is based on the result in (Yuan et al., 2020, Theorem 1), and the CONFET aims to model the probability density function $p(\bf{x}) = \lambda(x) / \Lambda$ directly, instead of handling Poisson process intensity $\lambda(\bf{x})$. I agree that this option is advantageous to avoid the integral in the likelihood (Eq. (2)) and to utilize rich density estimation methods. However, the aim of Poisson process modeling is to discover the distribution of the number of points and the distribution of their locations, as the authors themselves mentioned. In the CONFET, the distribution of the number, $n$, of points are assumed to be a Poisson distribution with mean $\Lambda$, that is, $n \sim {\rm Pois(\Lambda)}$ (Is that correct?); it is a simplistic assumption. In order to learn the complex Poisson process models, I think that it would be essential either to model the flexible mean $\Lambda$ or to handle the standard likelihood (Eq. (2)) as in Cox processes.

Also, the evaluation in Section 6 is for the probability density estimation methods. If the authors state that this work is the proposal of the new Poisson process models, the model comparison should be conducted with consideration of the estimation performance of $n$. In that case, it would be better to add the comparison with the recently-published Cox process models (e.g., [R1]). Another option is to place this work as the method of determining point locations given the number of samples $n$.

Could you please address and clarify the above concern?

[R1] C. Lloyd, T. Gunter, M. A. Osborne, and S. J. Roberts. Variational inference for Gaussian process modulated poisson processes. ICML 2015, 2015.

---

> ### Author Response · Authors · 2020-11-19
> **Response to Review 1**
>
> We want to thank the reviewer for valuable feedback.
>
> *The aim of point process modeling is to discover $p(n)$ and $p(x)$*
>
> As we already discussed in the paper and in Appendix A.1, we can model general point processes by modeling the distribution of the number of the points and the distribution of their locations, without losing generality. Not modeling the intensity $\lambda(x)$ directly does not mean we are unable to capture both of these distributions. Of course, we can trivially extend $p(n)$ beyond Poisson to get more expressive distributions. We did not find the need to do that in our work considering the empirical distribution of $p(n)$ in our data (Figure 7 in Appendix). Here, we argue that the biggest difficulty in modeling point processes is actually modeling locations given the number of points. We believe that this different perspective is very valuable to the point process community because we can use rich density estimation techniques to obtain strictly more expressive models (compared to e.g., inhomogeneous process). To summarize, we do not see the benefit of parametrizing $\lambda(x)$ compared to $p(x)$, but the obvious downside to $\lambda(x)$ is having to calculate the normalization constant.
>
> *Comparison to Cox process*
>
> In literature, there were two traditional ways to model point processes with clustering behavior: defining pseudolikelihood (Besag, 1975) or using a Cox process. Both of them usually struggle with the normalization term. Our model has the same expressive power as these methods but also has a closed-form likelihood. We did not initially compare to Cox process models since we feel this work is orthogonal to ours. That is, these approaches can be incorporated in our model easily, e.g. we could have Gaussian process modulated base density from which we sample points and transform them with a normalizing flow. As long as we can evaluate the base density likelihood and sample from it, nothing significantly changes in our model.
>
> In the updated version (see Section 6) we compare to (Lloyd et al., 2015) and demonstrate how our method can match its performance. Further, our model can provide stochastic intensities directly, without the expensive Gaussian process modulation.
>
> We conclude that our method offers a different perspective on point processes where the focus is on modeling locations without losing the expressiveness because we still model the whole point process (number of points + locations). We show how it outperforms other models for location modeling and other point process models (Lloyd et al., 2015).
>
> References:
> Besag, Statistical analysis of non-lattice data, Journal of the Royal Statistical Society (1975)
> Adams et al., Tractable Nonparametric Bayesian Inference in Poisson Processes with Gaussian Process Intensities, ICML (2009)
> Lloyd et al., Variational inference for Gaussian process modulated poisson processes, ICML (2015)

---

> > ### Comment · AnonReviewer1 · 2020-11-22
> > **Reply**
> >
> > I appreciate the authors' feedback.
> >
> > I understood the paper's place in the literature. The Poisson process formulation the authors adopted is certainly interesting and a good research direction in that one can utilize rich density estimation methods. But, the proposed method then seems somewhat incremental (as the other reviewers indicated): It is a good implementation of the Poisson process model based on continuous normalizing flows. I think this is a good job, but this might not ready for publication in terms of novelty.
> >
> > Thank you for adding the experiments. It would be more helpful to compare it with Cox-processes using test log-likelihood.

---

### Official Review · AnonReviewer3 · 2020-10-28
**review for Equivariant Normalizing Flows for Point Processes and Sets**

**Rating:** 6
**Confidence:** 3

**Review:**

The paper introduces a tractable likelihood model for point processes named Confet. The paper combines continuous time normalizing flows with graph-network equivariant transformations. They introduce some variations (stochastic/fixed/exact) with different approaches to compute the trace of the Jacobian.


Strengths:
The paper is well-written and introduces the subject really well. The background material is covered extensively and makes the paper pleasant to read. The illustrations are used nicely to illustrate concepts. The variations of the methods (stochastic/fixed/exact) is a nice addition in the context of point process modelling.


Weaknesses:
My main concerns stem from similarities between the proposed methods and existing methods in literature. The paper is correct that Eq. (8) is more expressive than the transformation in Kohler et al, but the reason for this is that Kohler et al. enforced an additional equivariance constraint on physical geometry. It is somewhat straightforward that if this constraint is alleviated, the functions g and h can be a more expressive neural networks. Further, there seem to be connections with graph normalizing flows that are not explored, as point clouds can be interpreted as graphs with edges between all points. Examples of works in this area are (Liu, et al. "Graph normalizing flows." 2019) and (Shi, et al. "GraphAF: a flow-based autoregressive model for molecular graph generation." 2020). Further, in the experimental section it would be good to compare against the method proposed by Kohler et al due to the similarities. In addition, I would like to request the authors to clarify and disentangle their work from Graph Flow literature and from the paper by Kohler et al..


Final comments:
I would like to again thank the authors for their time and responses. My concerns regarding novelty remain as described below, but the authors did clarify some aspects. For these reasons, I have raised my score from 5 to 6. If the work is not accepted at this venue, I would like to encourage the authors to continue with their work and submit to a later venue.

---

> ### Author Response · Authors · 2020-11-19
> **Response to Review 3**
>
> We want to thank the reviewer for the valuable feedback.
>
> *Further comparison to Koehler et al. (2020)*
>
> As we already pointed out in our paper, our model differs from (Koehler et al., 2020) by having a more expressive equivariant transformation and a new way to obtain the closed-form trace. The fact that function $h$ and $g$ (eq. 8) can be extended to be more expressive is known from (Zaheer et al., 2017). Koehler et al. (2020) propose a much simpler function with a closed-form trace to satisfy their symmetries. However, we use Theorem 1 from (Maron et al., 2020) that states that any G-equivariant layer is of the form as in our eq. 8. Therefore, we utilize the decoupling to obtain the closed-form trace in strictly more expressive networks.
>
> For completeness, in the updated version we compare to (Koehler et al., 2020), see Appendix C.2, and conclude that our method performs much better.
>
> *Comparison to graph methods*
>
> We are aware of the graph generation literature. For example, we cite (You et al., 2018) that use a similar autoregressive method to (Shi et al., 2020) on the breadth-first search ordering. In contrast, our baseline autoregressive model is working on the spatial point ordering. To implement the graph methods we would need to construct a fully connected graph with $O(n^2)$ edges for $n$ points. We would then need to define some canonical ordering (like BFS) on this graph. We do not see the benefit of using this approach compared to a more natural spatial ordering in our baseline. It may be the case that graph methods are well suited to some types of data, such as molecules, but in our case we have sets of points, without any underlying graph structure.
>
> Liu et al. (2019) propose an architecture extension of the Real NVP model using graph neural networks, with the split in the feature space. In our case, the features are spatial locations, therefore, this model would be equivalent to our inhomogeneous process baseline. It is strictly less expressive than our final model.
>
> References:
> Zaheer et al., Deep Sets, NeurIPS (2017)
> Maron et al., On Learning Sets of Symmetric Elements, ICML (2020)
> You et al., GraphRNN: Generating Realistic Graphs with Deep Auto-regressive Models, ICML (2018)
> Shi et al., GraphAF: A Flow-based Autoregressive Model for Molecular Graph Generation, ICLR (2020)
> Liu et al., Graph Normalizing Flows, (2019)

---

> > ### Comment · AnonReviewer3 · 2020-11-21
> > **Reply**
> >
> > I would like to thank the authors for their clarifications. After their rebuttal, two issues remain:
> >
> > The authors state that their method is more expressive than Kohler at al.. This was never disputed, my claim was that Kohler et al enforce an additional se(3) symmetry. It is not surprising that alleviating this constraint results in a more expressive model if se(3) symmetry undesired.
> > Secondly, in Liu et al. there are multiple versions of their model. One version has a graph structure where all points are connected and the adjacency matrix is not modelled. This situations is equivalent to modelling a point cloud. The difference to this model is than to utilise NODE/FFJORD-type transformations instead of RealNVP-type layers.
> >
> > I think the direction of this paper is very interesting but I find this paper somewhat incremental in terms of novelty because of the reasons above. I agree that the zero-trace method is interesting and novel, but this is not enough to change my mind regarding the novelty.

---

> > > ### Author Response · Authors · 2020-11-21
> > > **Thank you for your reply**
> > >
> > > Thank you for your reply.
> > >
> > > We agree that Koehler et al. have an inherently less expressive model and alleviating their constraints leads to a more expressive one. However, it is also true that naively doing this would lead to unstable training. Our novel method solves this and realizes both expressiveness and stable training with fast inference.
> > >
> > > Regarding the second point, we would again like to point out that Liu et al. is strictly less expressive than our model, equivalent to the inhomogeneous Poisson baseline. In contrast, our novel method is able to model interactions between the points and obtains closed form likelihood. We believe that the findings in our paper would benefit  the graph generation community too, because their methods so far relied on canonical orderings or simple densities (e.g. Liu et al).

---

### Official Review · AnonReviewer2 · 2020-10-28
**Interesting application of CNFs to point processes, but perhaps incremental**

**Rating:** 5
**Confidence:** 3

**Review:**

This paper proposes a method for modeling exchangeable sets of data, or point processes. Specifically, the paper is interested in applying normalizing flow methods to these point processes. The paper proposes a method using continuous normalizing flows, and compares the performance of their proposed method, Confet, to baselines on simulated and real data. The paper finds that Confet outperforms the other methods in terms of test loss.

The specific method proposed by the paper relies on continuous normalizing flows. These methods take a sample from a simple base distribution, and transform these samples to a more complicated distribution whose density can be calculated by solving an ODE. This method requires specifying a transformation $f$ whose trace Jacobian can be computed efficiently. In the case of point processes, $f$ needs to be _equivariant_, meaning that permuted inputs to $f$ must have the same output up to the same permutation. The authors use neural networks with equivariant layers, and propose three methods for computing the trace: a MC estimate using Hutchinson's estimator, fixing $f$ so the trace is 0, and exact calculation. In experiments, they find that the exact and stochastic computations perform well.

Point process models are an important area, and this paper does a nice job combining two fields: set modeling and normalizing flows. The paper is clear throughout, and the proposed method is simple and well-motivated. It's also nice that the paper provides three different trace estimators and compares their qualities.

My main criticism is that this paper describes an incremental improvement. It is not a new idea to use normalizing flows for modeling exchangeable data -- for example, the authors cite work by Bender et al, which also uses equivariant normalizing flows for exchangeable data. There is not much novelty with the proposed inference method; Hutchinson's estimator and exact trace computation are two common ways to train continuous normalizing flows.

It is not inherently a problem to me that the proposed method only makes incremental changes from existing methods; if experiments are convincing of the proposed method's performance, it would be a reason to accept. However I am not convinced by the current set of experiments. They compare to three baselines -- autoregressive, IHP, and IWAE -- which all consist of some kind of normalizing flow. Since the experiments only compare to baselines that use a normalizing flow for point process models -- rather than the most common point process models -- I'm not sure which hypothesis these experiments are trying to test. Is it that continuous normalizing flows are more appropriate than non-continuous normalizing flows in this context? This doesn't seem as useful to me as an experiment that would show that continuous normalizing flows outperform common and non-flow based models like DeepSets or Set Transformer.

Additionally, the experiments would be more convincing if there was a more extensive set of real-world datasets. The experiments are on two check-in datasets and one crimes dataset -- all location data. There exist a variety of point process/set data sets, such as point clouds, brain imaging data, and image anomaly detection. Because the real-world experiments are only on one kind of dataset, it's not convincing that the proposed model is useful for general sets and point processes.

In conclusion, I think the proposed idea is interesting and could be useful. However, it is incremental, and the current experiments do not sufficiently show that the model is important.

Pros:
 - straightforward idea
 - paper does a good job combining two fields: point processes and normalizing flows
 - clearly written and model is reproducible

Cons:
 - proposed method is incremental
 - insufficient experiments

---

> ### Author Response · Authors · 2020-11-19
> **Response to Review 2**
>
> We want to thank the reviewer for the valuable feedback.
>
> *Compare the novelty to other exchangeable flows, e.g. Bender at al. (2020)*
>
> In our paper we point out the issues in the existing works for exchangeable data and propose an elegant solution. In particular, Bender et al. (2020) impose a canonical ordering on the points and use an autoregressive flow which performs worse than our model in the experiments. Additional drawback is that sampling is sequential and slow. Our method does not need to order the points but can obtain the exact likelihood with an elegant model that allows interactions, and with straightforward sampling. This is a non-trivial improvement to previous works.
>
> *The novelty of the inference method*
>
> The exact trace computation can be easily obtained from the decoupling we define in eq. 8. Previously, Koehler et al. (2020) used a simple kernel based function which has closed form derivative, but we demonstrate our function outperforms it by a large margin (see our updated Appendix C.2). Before that, Chen and Duvenaud (2019) used closed form solution for general continuous normalizing flows. Our usage of *twice decoupling* within-point and between-point interaction allows us to have closed form trace, either by having it fixed or calculating only the within-point part, which has not been investigated previously. This allows us to have smooth training curves and orders of magnitude faster exact likelihood estimation (e.g. for anomaly detection), while retaining the expressive power to model complex processes.
>
> *All baselines consist of some kind of normalizing flow*
>
> First, we want to point out that we compare to the strongest baselines for exchangeable density modeling like Bender et al. (2020) and variational autoencoders. Our VAE baseline is novel in the context of point processes, closest related work is (Yuan et al., 2020) that defines the non-parametric model without the ability to sample completely new realizations. By adding a normalizing flow we make all of the baselines more expressive than they were previously, to make a fair comparison with our model.
>
> The question is not whether continuous normalizing flows are the best flows, but rather which approach: canonical ordering, latent model, or our closed-form density is the best. We show in the experiments that having a closed-form density for point processes is not only possible (compare to pseudolikelihood methods (Besag, 1975) that are still widely used today) but also performs the best.
>
> *Comparison to non-flow based models like Deep Sets or Set Transformer*
>
> There seems to be some kind of misunderstanding, our model already uses Deep Sets and Set Transformers for the transformations. Therefore, it inherits their expressive power. Further, since our model is a generative model, all the baselines should have a way of expressing the density or at least be able to sample new realizations. We extend the basic Deep Sets and Set Transformers with a VAE model and outperform it in the experiments. Having a normalizing flow in VAE model only gives it more modeling power, making it a very strong baseline that we then outperform.
>
> *More extensive set of real-world datasets*
>
> We used traditional point process datasets that are commonly used in the literature but we agree that a more extensive testing with other datasets would help confirm our method is relevant for set modeling in general. Because of that we include initial results on point cloud and discretized digits in Section 6.5. We will include further experiments for the camera-ready version.
>
> As requested by Reviewer 1, we additionally compare our model to Gaussian modulated point processes (Lloyd et al., 2015) -- a direct point process competitor, and show how we can match the performance and draw stochastic intensities directly from our model without the expensive Gaussian process modulation (see updated Section 6).
>
> References:
> Bender et al., Exchangeable Generative Models with Flow Scans, AAAI (2020)
> Koehler et al., Equivariant Flows: Exact Likelihood Generative Learning for Symmetric Densities, ICML (2020)
> Chen and Duvenaud, Neural Networks with Cheap Differential Operators, NeurIPS (2019)
> Yuan et al., Variational Autoencoders for Highly Multivariate Spatial Point Processes Intensities, ICLR (2020)
> Lloyd et al., Variational inference for Gaussian process modulated poisson processes, ICML (2015)

---

> > ### Comment · AnonReviewer2 · 2020-11-23
> > **Thanks for response**
> >
> > I want to thank the authors for responding and addressing my main points. I am more satisfied by the baselines and comparisons because of their response. I think this is interesting work but I still think it's incremental, even with the exact trace computation. I think the paper would be stronger if there was more explanation of how it differs from existing work, and if it included a more thorough set of experiments. I wouldn't be opposed to accepting it, but it seems like the other reviewers have the same concern about the work being incremental.

---

### Author Response · Authors · 2020-11-24
**General response**

We thank the reviewers for their valuable feedback.

In the discussion phase, we answered the questions that were raised, revised the draft, and implemented additional experiments. In particular, we compared to another equivariant normalizing flow (Koehler et al., 2020), cox process (Lloyd et al., 2015), and added point cloud and digit experiments. We showed that our method outperforms the competitors and is able to model challenging datasets.

Once again, we would like to point out the goals and scope of this paper:
- In this paper we propose an alternative perspective to modeling point processes where the focus is on modeling the $p({x_1, ..., x_n})$ and $p(n)$ separately, without losing generality. This allows us to have a closed form likelihood, in contrast to modeling intensity $\lambda(x)$.
- By modeling $p({ x_1, ..., x_n })$ with a continuous normalizing flow we allow interactions between points getting a strictly more expressive set of densities. It also does not impose any additional restrictions like competing models for exchangeable data (e.g. Bender et al., 2020).
- We propose a novel decoupling method for permutation equivariant transformations that gives a closed form trace which allows more stable training and much faster inference. The implementation of decoupling in code is straightforward and can be combined with general decoupling methods (Chen and Duvenaud, 2020).
- We show that our model outperforms competitors across the variety of real-world and synthetic datasets, achieving better likelihood and higher quality samples.

References:
Lloyd et al., Variational inference for Gaussian process modulated Poisson processes, ICML (2015)
Chen and Duvenaud, Neural Networks with Cheap Differential Operators, NeurIPS (2019)
Koehler et al., Equivariant Flows: Exact Likelihood Generative Learning for Symmetric Densities, ICML (2020)
Bender et al., Exchangeable Generative Models with Flow Scans, AAAI (2020)

---

### Decision · Program_Chairs · 2021-01-07
**Final Decision**

**Decision:**

Reject

**Comment:**

The authors model point processes with equivariant normalizing flows. Reviewers agreed that the paper is well written and addresses a problem of interest to the ICLR community, some reviewers considered the contribution to be incremental.  Perhaps the biggest contribution is a closed form expression for the trace that needs to be computed as part of the normalizing flow, which is valuable but not particularly emphasized. The authors combine this trace formulation with an equivariant normalizing flow to model the conditional density of point locations given cardinality. (As an aside, it was unclear to me if and how those conditional distributions share parameters; in some contexts, the conditional density could look very different depending on the number of points in the set.) Overall, the paper is interesting but needs a little more to lift it over the bar.